# Epidemiology of Fungal Colonization in Children Treated at the Department of Oncology and Hematology: Single-Center Experience

**DOI:** 10.3390/ijerph19042485

**Published:** 2022-02-21

**Authors:** Joanna Klepacka, Zuzanna Zakrzewska, Małgorzata Czogała, Magdalena Wojtaszek-Główka, Emil Krzysztofik, Wojciech Czogała, Szymon Skoczeń

**Affiliations:** 1Department of Microbiology, University Children’s Hospital, Wielicka 265, 30-663 Krakow, Poland; jklepacka@usdk.pl (J.K.); wojciech.czogala@uj.edu.pl (W.C.); 2Department of Oncology and Hematology, University Children’s Hospital, Wielicka 265, 30-663 Krakow, Poland; zuzanna.zakrzewska@doctoral.uj.edu.pl (Z.Z.); malgorzata.czogala@uj.edu.pl (M.C.); 3Department of Pediatric Oncology and Hematology, Institute of Pediatrics, Jagiellonian University Medical College, 30-663 Krakow, Poland; 4Student Scientific Group of Pediatric Oncology and Hematology, Jagiellonian University Medical College, 30-663 Krakow, Poland; magda.wojtaszek-glowka@student.uj.edu.pl (M.W.-G.); emilkrzysztofik.ek@gmail.com (E.K.)

**Keywords:** fungal infection, mycology, hematology, pediatric oncology, yeast, *Candida* spp., *Candida albicans*, candidiasis

## Abstract

Oncological patients are especially predisposed to fungal infections due to multiple risk factors and immunocompromising treatment. Epidemiological research regarding pediatric oncologic patients is still insufficient, and existing data are difficult to generalize on different populations. Therefore, we aimed to analyze fungal infections and fungal epidemiology in the Department of Oncology and Hematology of the University Children’s Hospital in Krakow with help from the Clinical Microbiology Department. During the chosen period of 2005 and 2015–2020, 2342 tests were performed in our ward on 847 patients. Analyzed samples were divided into five source groups. The amount of patients with positive test results was 62.5%. The year with the highest detection level was 2005. The most frequent pathogen was *Candida albicans*, with a significant decrease in tendency. An increase in non-albicans species was observed. *Candida parapsilosis* was not frequently observed compared to similar studies. We noticed an increase in positive results from the urinary tract material. Our results confirmed that fungal infections are still an issue, and they may indicate the efficacy of prophylaxis. The majority of our results are consistent with the literature, yet we managed to emphasize data unique to our patients’ population. Our findings are helpful in clinical work and for further studies in our center.

## 1. Introduction

Infections are a major issue in the treatment of immunocompromised pediatric patients, as they are a significant cause of morbidity and mortality. Oncological patients are at a high risk of developing complicated infections that are difficult to treat [1,2,3]. Cancer is a state of impaired innate and adaptive immunity due to both treatment effects and the malignancy itself, and it leads to increased susceptibility to infections. Additional factors contributing to a patient’s vulnerability are, among others, prolonged leukopenia and neutropenia, malnutrition, disruption of physiological barriers of the mucosa and the skin, and impairment of microbiota or instilment of foreign bodies such as a central venous catheter. Early diagnosis and treatment are important since delays make the infectious process progress quickly, or even generalize. Therefore, prophylaxis, rapid detection, and effective treatment are important factors for overall survival [1].

Fungal infections are not as common as those caused by bacteria or viruses, but they are insidious and can cause severe complications, especially in patients with impaired immunity [1,4,5]. Clinical manifestations of infection may vary from a superficial or cutaneous form to a life-threatening, systemic disease [5,6,7]. *Candida* spp. and *Aspergillus* spp. are frequently isolated from immunocompromised patients [5,8,9]. The most common fungi causing sepsis during oncologic treatment are: *Candida* spp., *Aspergillus* spp., *Zygomycetes*, *Cryptococci*, and *Pneumocystis jiroveci* [1]. Mycological identification of pathogens is challenging due to problems with obtaining adequate material, as well as culturing itself. Furthermore, isolation of fungi from clinical material may indicate not only an infection or a disease, but also colonization. Interpretation of results is therefore difficult [5]. In addition, currently widely used anti-fungal prophylaxis can further restrict the possibility of an accurate infection diagnosis. All of the above impede evaluation of the true incidence of fungal infections.

The majority of epidemiological data on this topic has come from case studies or single-center studies concerning a single fungal species or examining specific populations [10,11,12,13,14,15,16]. Many researchers have investigated a single fungal species in studies published over 10 years ago [17,18,19,20,21]. Furthermore, the knowledge of etiological factors, both in general and typical for clinical facilities, is crucial for focused therapy adjustment. The need for more data in the pediatric population is still valid.

We aimed to retrospectively analyze fungal infections and fungal diagnostics in the pediatric hemato-oncologic patients of the University Children’s Hospital in Krakow (UCH) during the period of 2015–2020 from a mycological point of view. We referred to historical data from the randomly selected year of 2005 in order to observe whether any change in incidence appeared.

## 2. Materials and Methods

The Clinical Microbiology Department of UCH performed a total of 23,334 mycological diagnostic tests in the year 2005 and during the years 2015–2020, which 2342 (10%) were run for the Department of Oncology and Hematology (DOH). Diagnostics involved mycological and serological testing. The main focus was to analyze the incidence of test results during the period of 2015–2020. The year 2005 was randomly selected as a historical time reference to note possible changes.

Every child admitted to the DOH who underwent mycological testing was included in the analysis, constituting a group of 847 patients (472 male). The population included children with various hemato-oncologic disorders. Samples were collected when there were clinical indications. In our ward, diagnostics were performed in patients with signs of infection. Mycological samples were collected from adequate sources if there was a possibility of fungal etiology. Prophylaxis was used according to the national and EUCAST guidelines. We were solely interested in the amount and results of mycological testing performed by our department; therefore, we did not analyze detailed clinical data.

The number of patients through the years was as follows: 2005, 111; 2015, 133; 2016, 114; 2017, 119; 2018, 138; 2019, 132; and 2020, 100. The year 2005 was set as a baseline to estimate changes in the amounts of performed tests and cultured yeast fungi.

Analyzed microbiological samples were divided into 5 groups based on source: (1) gastrointestinal tract (stool or anal swabs), (2) upper respiratory tract (nasopharyngeal, nasal, or throat swabs), (3) lower respiratory tract (bronchial lavage, bronchoalveolar lavage, intrapleural fluid, or aspirates), (4) urinary tract (urine), and (5) other (collected from skin lesions, urethral catheter, vaginal smears, eye swabs, or wound swabs).

In addition, serological fungal tests were conducted for 392 patients (224 male), with 63 tests in 2015, 68 in 2016, 58 in 2017, 67 in 2018, 68 in 2019, and 68 in 2020. Serological tests were not performed in 2005. Detailed information concerning the performed tests can be found in Table 1.

Mycological testing included: 1. direct and stained samples; 2. liquid surface culture possibly with Sabourauda agar culture with or without antibiotics in the temperature of 30 °C; and 3. identification of cultivated yeasts. Identification (no. 3) included: a. filamentation (germ tube test); b. differential surface culture (corn meal agar, differentiating presence or lack of chlamydospores and possibly pseudomycelium), c. pre-prepared biochemical tests ID 32 C (bioMerieux) (bioMérieux SSC Europe Sp. z o. o., Poland) and BD Phoenix Yeast (Becton, Dickinson and Company), and d. mass spectrometry (MALDI-TOF MS) using matrix-assisted desorption/laser ionization time-of-flight.

Serological diagnostics of fungal infections consisted of: 1. anti-mannan Candida antibodies in serum or plasma (Platelia Candida Ab Plus, Bio-Rad Laboratories, Inc.), 2. IgG anti-Aspergillus antibodies in serum or plasma (Platelia Aspergillus IgG, Bio-Rad Laboratories, Inc.), 3. Candida mannan antigen in serum or plasma (Platelia Candida Ag Plus, Bio-Rad Laboratories, Inc.), and 4. galactomannan Aspergillus antigen in serum or plasma (Platelia Aspergillus Ag, Bio-Rad Laboratories, Inc.)

For statistical analysis of the collected data, we used Statistica 13.3 (StatSoft, Tulsa, OK, USA). The incidence of positive results for each year was compared using Pearson’s chi-squared test. Our main focus was to compare each year from the 2015–2020 period with 2005, in addition to conducting a year-by-year comparison.

## 3. Results

During the chosen period, 1035/2342 mycological tests were positive. Figure 1 shows the distribution of the mycological tests collected from DOH patients during the studied years, including the number of positive results.

During the studied period, at least one positive result of mycological tests was found in 529 of 847 patients (62.5%). Details are shown in Figure 2.

*Candida albicans* was the most frequently isolated yeast species each year.

The prevalence of positive *Candida albicans* results in the studied years was significantly higher in 2005 (65%, 105/161) than in the following years: 2015, 42% (58/139, *p* = 0.00001); 2016, 45% (70/156, *p* = 0.0003); 2017, 51% (70/138, *p* = 0.0052); 2018, 39% (76/193, *p* = 0.00001); 2019, 43% (66/153, *p* = 0.001); and 2020, 52% (49/95, *p* = 0.013).

A significant increase in positive cultures of *Candida glabrata, Candida krusei, Candida lusitaniae*, and *Saccharomyces* was observed between the analyzed years of 2015–2020 and 2005, with the following details: An increase in *Candida glabrata* cultures compared to 2005 (6.8%, 11/161 patients) was observed in the following years: 2015, 17% (24/139, *p* = 0.005); 2016, 21% (33/156, *p* = 0.002); 2017, 15% (21138, *p* = 0.0194); 2018, 18.6% (36/193, *p* = 0.011); 2019, 10% (16/153, *p* = 0.25); and 2020, 9% (9/95, *p* = 0.45).

An increase in cultures of *Candida krusei* was found in the years 2015 (8.6%, 12/139 patients, *p* = 0.001), 2016 (16%, 25/156, *p* = 0.00001), 2017 (9%, 13/138, *p* = 0.0001), and 2018 (11%, 22/193, *p* = 0.00001), compared to 2005 (0%, 0/161). In the years 2019 and 2020, the percentages of positive cultures were lower with no differences from 2005: 2019, 1.3% (2/153, *p* = 0.1456); and 2020, 3% (3/95, *p* = 0.95).

A significant increase in cultures of *Candida lusitaniae* compared to 2005 (1.8%, 3/161 patients) was observed in 2016 (9.6%, 15/156, *p* = 0.0029) and 2018 (7%, 14/193, *p* = 0.0182). The percentages of *Candida lusitaniae* positive results in the years 2015 (3.6%, 5/139), 2017 (0%, 0/138), 2019 (4.6%, 7/153), and 2020 (4%, 4/95) did not differ from the year 2005.

An increase in cultures of *Saccharomyces* was observed compared to 2005 (2.5%, 4/161 patients) in the following years: 2015, 10% (14/139, *p* = 0.058); 2016, 5.8% (9/156, *p* = 0.14); 2017, 10.9% (15/138, *p* = 0.030); 2018, 10% (19/193, *p* = 0.0052); 2019, 7.8% (12/153, *p* = 0.0309); and 2020, 12.6% (12/95, *p* = 0.0012).

Remaining species did not exceed 11% of all cultures. Details can be found in Figure 3.

We additionally analyzed the collected data divided into the sources of examined clinical samples, which can be found below.

Figure 4 shows the number of collected samples each year divided into five source groups: (1) gastrointestinal track, 1668; (2) upper respiratory tract, 66; (3) lower respiratory tract, 134; (4) urinary tract, 261; and (5) other tests, 261.

Analysis of 1668 gastrointestinal tract samples collected from 754 patients revealed the following results. The amount of patients with positive results was 75%: 2005 (66/88 patients); 2015, 67% (78/116); 2016, 65% (69/105); 2017, 68% (74/108); 2018, 68% (86/126); 2019, 59% (72/122); and 2020, 64% (57/89). No statistically significant differences in the proportions of patients with positive and negative results between the years were found. The most frequently isolated species in the gastrointestinal tract samples can be found in Figure 5.

A total of 66 samples from the upper respiratory tract were collected from 47 patients and revealed 24 different yeast isolates. More details are shown in Table 2.

The prevalence of patients with positive results in the studied years was as follows: 2005, 36% (5/14 patients); 2015, 33% (2/6); 2016, 50% (3/6); 2017, 40% (2/5); 2018, 30% (3/10); 2019, 75% (3/4), and 2020, 0% (2 patients). There were no statistically significant differences.

We collected 134 lower respiratory tract samples from 54 patients. Twenty-four different yeast species were isolated. More details are shown in Table 3.

The percentages of patients with positive results in the studied years were as follows: 2005, 25% (2/8 patients); 2015, 33% (2/6); 2016, 11% (1/9); 2017, 9% (1/11); 2018, 22% (2/9); 2019, 50% (3/6); and 2020, 20% (1/5). There were no statistically significant differences.

We collected 261 urinary tract samples from 167 patients with 45 yeast isolates detected. More details can be found in Table 4.

The prevalence of patients with positive results in the studied years was as follows: 2005, 22% (8/36 patients); 2015, 6% (2/31); 2016, 11% (2/18); 2017, 9% (2/22); 2018, 31% (6/19); 2019, 35% (7/20); 2020, 9% (2/21).

Significant differences were found between 2015 and 2018 (*p* = 0.0265), as well as 2019 (*p* = 0.013)

Additional serological diagnostic results including antibody levels and the presence of fungal antigens for *Aspergillus* and *Candida* in DOH patients are described in Table 5 and Table 6. No significant differences were found between the years.

## 4. Discussion

The most common pathogen detected in our patients was *Candida albicans*. Although this finding is consistent with previously published literature, [20,22,23,24,25] *Candida albicans* had a stronger adherence capacity during testing than other *Candida* species, [9] which could make it easier to detect. Regardless, a significantly higher amount was detected in 2005, with a decrease in subsequent years. Additionally, we observed a significant increase in the detection of *Candida glabrata* and *Saccharomyces* in subsequent years. *Candida krusei* had a significantly higher detection in 2015–2018. *Candida lusitaniae* detection was higher only in 2016 and 2018 compared to 2005. Nonetheless, the use of prophylactic antifungal drugs could potentially be the reason for the observed shift to non-albicans species [17,23,24,26].

A prolonged and repeated therapy with wide-spectral antibiotics or steroids, use of central venous catheters, and frequent mucositis are predisposing factors to fungal infections in children with hematologic and oncologic diseases [1,27,28]. Adequate antifungal prophylaxis is therefore introduced in these patients, particularly those at higher risk, e.g., children after stem cell transplantation or patients with prolonged neutropenia, according to national and EUCAST guidelines [1,5,22]. Although often necessary, longitudinal prophylaxis can result in drug resistance. An example is *Candida lusitaniae* resistance and the stimulation of its colonization after treatment with amphotericin B and decreased sensitivity to the drug [29,30]. Similar effects have been observed in a prophylactic use of fluconazole with *Candida glabrata* and *Candida krusei* colonization [31,32]. In addition, probiotics used in patients with immunosuppressive disorders have been observed to cause colonization, possibly with sepsis, with *Saccharomyces* [33,34,35].

In contrast to other centers, we detected fewer *Candida parapsilosis* isolates [11,17,24,36].

In general, the majority of patients had positive fungal tests results (62.5%), meaning the decision to diagnose was justified. In the described centers, fluconazole has been routinely used as prophylaxis in majority of cases. A decrease in *Candida albicans* detection suggests that an anti-fungal prophylaxis was successful. Other studies have confirmed a decrease in fungal infection prevalence [13,22,37,38]. In addition, the highest number of patients was admitted for treatment in 2018. Fewer patients were admitted to the Department in 2020 due to the COVID-19 pandemic, which might have influenced the diversity of the results. Despite the fact that the percentage of positive results this year was not significantly different, it might not be fully reliable.

We noticed a significant difference in positive fungal samples detected from the urinary tract between 2015 and 2018–2019. *Candida albicans* was the leading pathogen in this sample source, similar to other referred studies [11]. This observation is important, suggesting that urinary tract infections could end with fungemia [39,40], and need more inspection.

Our experience indicates that blood cultures and antigen testing do not add important information to fungal diagnostics. Colonization with fungal pathogens, *Candida albicans* in particular, is very common, making antigen testing less useful without information on clinical presentation. Our results showed that serological detection was very limited. Nevertheless, if an increase in antibody levels was detected, it could suggest the status of fungal infection [41,42,43]. We suggest that detection of antigens can be more informative in DOH patients. Fungal antigens were detected more frequently than antibodies. Thus, antigen testing could be treated as a screening method in immunocompromised patients. Establishing a validated method for the detection of fungal antigens in clinical samples could be beneficial and helpful in predicting upcoming infections.

Several large studies on invasive fungal infections have been previously conducted [44,45,46] with the use of EORTC-IFG/NIAID-MSG classification criteria [47], which we did not use due to a lack of clinical data. Our study focused on fungal epidemiology in DOH patients mainly from a mycological point of view. This could be regarded as a limitation, as we could not be certain whether patients dealt with an infection or colonization. Nonetheless, the samples were collected from cases with occurring symptoms. In addition, the analyzed data came from a single center of a large, tertiary care pediatric referral hospital. Despite the limitations, we consider the presented data useful and a good introduction to further studies, as it is the first study presenting fungal testing results from our center. The knowledge of the particular epidemiology in our center is important for making therapeutic decisions regarding antifungal medication. Existing data from other centers should be carefully generalized, since there may be differences in patient population, risk factors, and treatment demeanor. Regardless, the prevalence of fungal colonization in oncological and hematological patients confirmed the importance of implementing prophylaxis in immunocompromised children with a high risk of infection in order to prevent generalized opportunistic infection.

## 5. Conclusions

The majority of patients who qualified for mycological diagnostics each year had positive test results, indicating that fungal infections are still a concern for our patients. Nonetheless, a significant decrease in *Candida albicans* compared the benchmark set at the year 2005, as well as an increase in non-albicans species, suggested that the cause was prophylaxis, despite the fact that *Candida albicans* was still the most frequently detected culture. Contrary to similar studies, we detected a low amount of *Candida parapsilosis* isolates. The differences found in the urinary tract material were significant and need more insight. We concluded that our analysis provides good suggestions for clinical work and a baseline for further research regarding this topic, emphasizing the data unique to our center.

## Figures and Tables

**Figure 1 ijerph-19-02485-f001:**
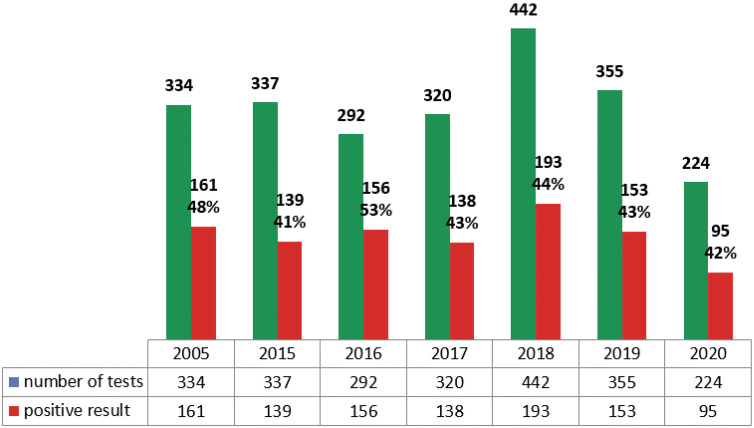
Mycological tests performed in DOH in the studied years, including positive results.

**Figure 2 ijerph-19-02485-f002:**
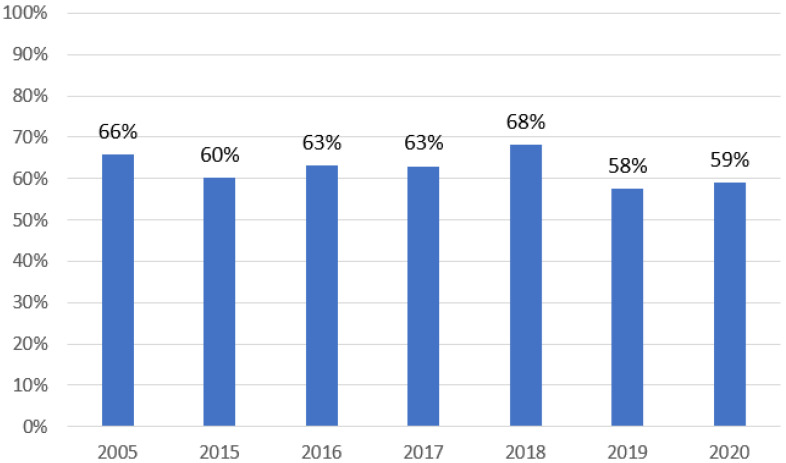
Prevalence of patients with positive mycological tests in the studied years.

**Figure 3 ijerph-19-02485-f003:**
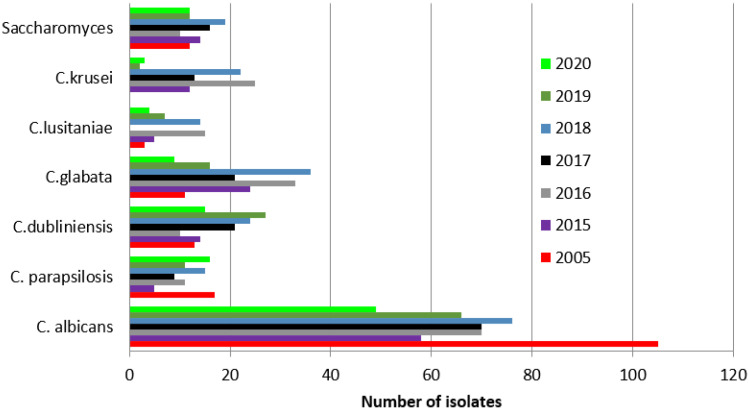
Most frequently isolated yeast species and *Saccharomyces* from clinical samples each year.

**Figure 4 ijerph-19-02485-f004:**
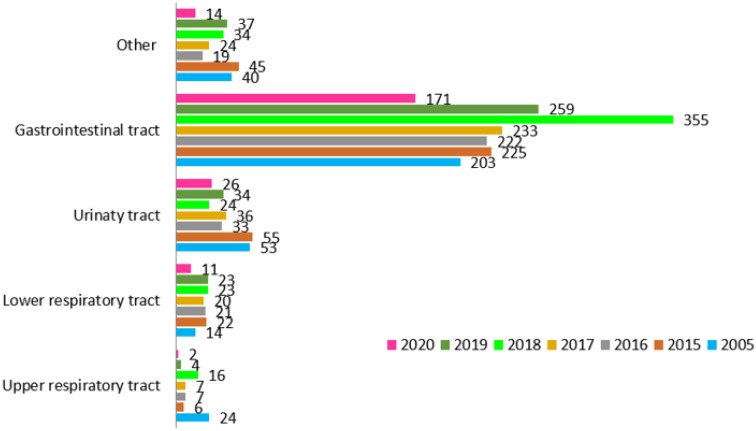
Number of mycological samples divided into 5 source groups.

**Figure 5 ijerph-19-02485-f005:**
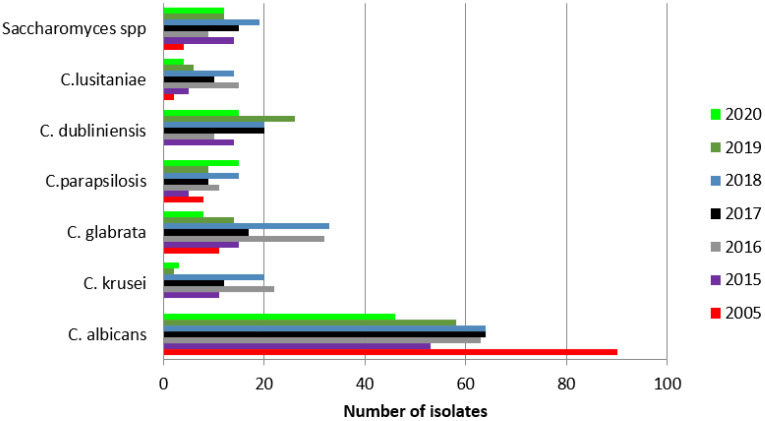
Yeast species and *Saccharomyces* isolated from the gastrointestinal tract in the studied years.

**Table 1 ijerph-19-02485-t001:** Patient and test numbers in the studied years.

	2005	2015	2016	2017	2018	2019	2020
Number of patients (cultures)	111	133	114	119	138	132	100
Male (%)	69 (62.16)	60 (45.11)	62 (54.36)	68 (57.14)	76 (55.1)	52 (39.4)	57 (57)
Number of mycological tests	334	337	292	320	442	355	234
Number of patients (serology)	0	63	68	58	67	68	68
Male (%)	0	30 (47.62)	36 (52.94)	39 (67.24)	40 (59.7)	41 (60.29)	38 (55.88)
Number of serological tests	0	494	497	344	584	884	641

**Table 2 ijerph-19-02485-t002:** Yeast species isolated from the upper respiratory tract in the studied years.

Year (Amount of Tests)	2005 (24)	2015 (6)	2016 (7)	2017 (7)	2018 (16)	2019 (4)	2020 (2)
Species	Amount of Isolates (Percentage of Tests)						
*Candida albicans*	6 (25%)	1 (17%)	3 (43%)	0	2 (12%)	3 (75%)	0
*Candida glabrata*	1 (4%)	0	0	0	1 (6%)	0	0
*Candida parapsilosis*	1	0	0	0	0	0	0
*Candida krusei*	0	0	1 (14%)	0	0	0	0
*Candida dubliniensis*	0	0	1 (14%)	1 (14%)	0	0	0

**Table 3 ijerph-19-02485-t003:** Yeast species and *Saccharomyces* isolated from the lower respiratory tract in the studied years.

Year (Amount of Tests)	2005 (14)	2015 (22)	2016 (21)	2017 (20)	2018 (23)	2019 (24)	2020 (11)
Species	Amount of Isolates (Percentage of Tests)						
* Candida albicans *	1 (7%)	0	0	0	3 (13%)	0	1 (9%)
* Candida glabrata *	0	4 (18%)	0	0	1 (4%)	1 (4%)	0
* Candida krusei *	0	0	0	1 (5%)	0	0	0
*Candida kefyr*	3 (21%)	0	0	0	0	0	0
*Candida inconspicua*	3 (21%)	0	0	0	0	0	0
*Candida tropicalis*	0	0	0	0	0	1 (4%)	0
*Saccharomyces*	3 (21%)	0	1 (5%)	0	0	0	0

**Table 4 ijerph-19-02485-t004:** Yeast species isolated from the urinary tract in the studied years.

Year (Amount of Tests)	2005 (53)	2015 (55)	2016 (53)	2017 (36)	2018 (24)	2019 (34)	2020 (26)
Species	Amount of Isolates (Percentage of Tests)						
* Candida albicans *	6 (11%)	0	3 (6%)	4 (11%)	4 (17%)	2 (6%)	2 (8%)
* Candida glabrata *	4 (7%)	5 (9%)	1 (2%)	0	1 (4%)	1 (3%)	0
* Candida parapsilosis *	2 (4%)	0	0	0	1 (4%)	1 (3%)	1 (4%)
* Candida dubliniensis *	0	0	0	0	2 (8%)	2 (6%)	0
*Candida lusitaniae*	0	0	0	0	0	1 (3%)	0

**Table 5 ijerph-19-02485-t005:** Serological test results detailed by years.

Result/Year	2015	2016	2017	2018	2019	2020
**Aspergillus antigen (number of samples)**	129	144	92	157	220	157
Negative (index < 0.5)	123	140	91	155	218	139
Positive (index > 0.5)	6 (4.65%)	4 (2.78%)	1 (1.09%)	2 (1.27%)	2 (0.90%)	18 (11.47%)
**Anti-Aspergillus antibodies (number of samples)**	110	105	74	128	212	147
Negative (<5 AU/mL)	107	97	73	128	212	145
Intermediate (5–10 AU/mL)	2 (1.87%)	8 (8.25%)	1 (1.37%)	0	0	2 (1.38%)
Positive (>10 AU/mL)	1 (0.94%)	0	0	0	0	0
**Candida antigen (number of samples)**	135	141	97	162	229	167
Negative (<62.5 pg/mL)	125	137	95	157	214	159
Intermediate (62.5–125 pg/mL)	4 (3.20%)	1 (0.73%)	0	2 (1.32%)	9 (4.20%)	3 (1.89%)
Positive (>125 pg/mL)	6 (4.80%)	3 (2.19%)	2 (2.11%)	3 (1.91%)	6 (2.80%)	5 (3.15%)
**Anti-Candida antibodies (number of samples)**	120	101	80	137	220	166
Negative (<5 AU/mL)	73	69	54	98	132	73
Intermediate (5-10 AU/mL)	27 (22.50%)	19 (27.54%)	17 (31.48%)	20 (20.40%)	73 (55.30%)	52 (71.23%)
Positive (>10 AU/mL)	20 (27.40%)	13 (18.84%)	9 (16.67%)	19 (19.40%)	15 (11.36%)	41 (56.16%)

**Table 6 ijerph-19-02485-t006:** Patients with positive results of serological testing each year.

	2015	2016	2017	2018	2019	2020	*p*
**Aspergillus antigen**							0.675938
Number of patients	63	69	57	66	68	65	
Number of patients with a positive result (percentage)	2 (3.2%)	4 (5.8%)	1 (1.7%)	2 (3.0%)	2 (2.9%)	2 (3.1%)	
**Anti-Aspergillus antibodies**							0.698505
Number of patients	57	57	49	62	65	64	
Number of patients with a positive result (percentage)	1 (1,7%)	1 (1,7%)	0	0	0	1 (1.6%)	
**Candida antigen**							0.346278
Number of patients	63	69	57	67	68	68	
Number of patients with a positive result (percentage)	2 (3.2%)	1 (1.4%)	2 (3.5%)	3 (4.5%)	5 (7.3%)	6 (8.8%)	
**Anti-Candida antibodies**							0.531438
Number of patients	57	56	50	62	65	66	
Number of patients with a positive result (percentage)	9 (15.8%)	11 (19.6%)	7 (14.0%)	10 (16.1%)	8 (12.3%)	16 (24.2%)	

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
