# Peer review of "Epidemiology of Fungal Colonization in Children Treated at the Department of Oncology and Hematology: Single-Center Experience"

_ijerph, 2022, doi:10.3390/ijerph19042485_

Round 1
Reviewer 1 Report
the topic dealt with in the article is not very interesting as it is normal for immunocompromised patients to contract infectious diseases. it would be interesting to understand why the types of bacterial or fungal strains that patients come into contact with change over the years. Is a form of gene modification possible in the viral or fungal strain? has there been a form of drug resistance? Has there been a change in the chemotherapy treatment core that has led to the onset of these changes in infection by bacteria or fungi?
Furthermore, there are some character errors (eg candida and aspirgellus should be written in italics, like the other fungal strains mentioned). The tables and graphs are not very clear so they should be completely revised. Figure 2 is redundant because it is very similar to Figure 1 in concept.
The data analysis shown under Figure 2 is very detailed, perhaps it should be simplified and part of the comment on Figure 3 could be inserted.
The graphs must all be redesigned because they are unclear on a graphical level. on some of them the DS are missing.
Check the bibliographic references (eg ref. 44): some are old and something newer should be found.
Author Response
Dear reviewer, thank you for interesting comments.
- The objective of the study was solely analysis of number of mycological tests and its results. We did not present any data on drug resistance or genetic modifications of the microorganisms.
- We corrected some errors that you have noticed.

Reviewer 2 Report
Manuscript requires additional editing for grammar and readability plus there are some basic errors in scientific standards.
This work is quite specific to the locality where it was done so has limited comparability. The use of the 2005 timepoint may not add much to the findings as so many confounders may be affecting the changes seen between timepoints. Serological data does not add much so may not be useful in this manuscript. Makes a good argument for further investigation as this is clearly an exploratory piece of research so the conclusions need to be similarly tentative at this point.

Author Response
Dear reviewer, thank you for the interesting comments.
We edited the manuscript according to your suggestions.
The presented study concentrates solely on the numer of mycological tests and its results from the epidemiological point of view and it should be treated as exploratory piece of research.
Analysis of the historical data was performed year by year. The year 2005 was randomly selected as historical time reference to notice possible changes.

Round 2
Reviewer 1 Report
In table 5, numbers after the comma are not always two.
Names of bacterial strains are not always written in the same way. Sometimes they are written in italics, others in bold.
Check for some spelling and synthase errors.
Check the way of writing the contacts of the corresponding authors: they are written in different characters
Author Response
Dear Reviewer, thank you very much for your suggestions. We have changed the manuscript accordingly.
Best regards